# Tumor-Infiltrating Lymphocytes (TILs) in Breast Cancer: Prognostic and Predictive Significance across Molecular Subtypes

**DOI:** 10.3390/biomedicines12040763

**Published:** 2024-03-29

**Authors:** Aleksandra Ciarka, Michał Piątek, Rafał Pęksa, Michał Kunc, Elżbieta Senkus

**Affiliations:** 1Department of Pathomorphology, Medical University of Gdańsk, M. Skłodowskiej-Curie 3a, 80-214 Gdańsk, Polandmichal.kunc@gumed.edu.pl (M.K.); 2Department of Oncology, Institute of Medical Sciences, University of Opole, pl. Kopernika 11a, 45-040 Opole, Poland; 3Department of Oncology and Radiotherapy, Medical University of Gdańsk, M. Skłodowskiej-Curie 3a, 80-214 Gdansk, Poland

**Keywords:** breast cancer, tumor microenvironment, prognosis, cancer immunotherapy, predictive biomarkers, inflammation

## Abstract

Tumor-infiltrating lymphocytes (TILs) are pivotal in the immune response against breast cancer (BC), with their prognostic and predictive significance varying across BC subtypes. In triple-negative BC (TNBC), higher TIL levels correlate with improved prognosis and treatment response, guiding therapeutic strategies and potentially offering avenues for treatment de-escalation. In metastatic TNBC, TILs identify patients with enhanced immunotherapy response. HER2+ BC, similar to TNBC, exhibits positive correlations between TILs and treatment response, especially in neoadjuvant settings. Luminal BC generally has low TILs, with limited prognostic impact. Single hormone receptor-positive BCs show distinct TIL associations, emphasizing subtype-specific considerations. TILs in ductal carcinoma in situ (DCIS) display ambiguous prognostic significance, necessitating further investigation. Standardizing TIL assessment methods is crucial for unlocking their full potential as biomarkers, guiding treatment decisions, and enhancing patient care in BC.

## 1. Introduction

Tumor-infiltrating lymphocytes (TILs) are immune cells that infiltrate the tumor microenvironment and play a critical role in the immune response against cancer. Interactions between the cells that make up TILs are involved in the development of memory T cells and B cells that are specific to tumor antigens. TILs in breast cancer (BC) consist mainly of CD8+ cells, CD4+ cells, FOXP3+, and CD19+ cells, and less frequently CD56+ NK cells. The majority of TILs are located in the stromal region directly adjacent to the tumor, which are referred to as stromal TILs (sTILs). A smaller portion is found within the tumor itself, and they are thus termed intratumoral TILs (iTILs) [1]. The assessment of the amount of sTILs and iTILs may have prognostic and predictive value in BC, although the clinical implications reported for both types of TILs are not consistent [2].

The cellular composition of TILs in BC may be of predictive and prognostic significance. For example, a higher number of CD8+ cells in TILs before treatment and in the lymphoid infiltrate in the tumor bed after neoadjuvant treatment is associated with a higher incidence of pathological complete response (pCR) [3]. On the other hand, a higher number of FOXP3+ TILs is associated with shorter overall survival (OS) [4]. Moreover, the cellular composition of TILs may vary depending on the presence of certain mutations in BC cells. In *PIK3CA*-mutated ER+ BC, a higher number of CD8+ cells was observed and was associated with a higher risk of recurrence [5]. Other molecular mechanisms that regulate TILs in BC encompass immune checkpoint pathways. Immune checkpoint ligands like programmed death-ligand 1 (PD-L1) expressed by tumor cells can inhibit TIL activity and promote immune evasion [6]. Accordingly, TIL density displays potential predictive value in the context of immunotherapy [7]. TIL level and its composition may also be impacted by prior treatments, such as neoadjuvant therapy, which may cause an increase or decrease of TILs in residual disease, with usually a decreased number of FOXP3+ cells [3,8,9,10].

In BC, especially those with an extensive lymphocyte infiltrate, TILs can also manifest as so-called tertiary lymphoid structures (TLSs), which contain germinal centers and thus resemble lymph nodes in structure. Their presence, which may be related to higher concentrations of chemokine CXCL13 produced by CD4+ cells, is associated with better response to adjuvant chemotherapy (AChT) and longer disease-free survival (DFS) [11].

Lymphocytic infiltrates are found both in primary BCs and in BC metastases. Unlike primary tumors, metastases are usually characterized by a lower number of TILs, and the prognostic significance of TIL infiltration in metastatic foci is not well established and warrants further investigation [12].

In 2014, the International TILs Working Group took a significant step towards standardizing the assessment of TILs in BC by issuing comprehensive recommendations [1]. These guidelines aimed to ensure accurate and reproducible evaluation of TILs, emphasizing the importance of a standardized methodology. To promote consistency across different research and clinical settings, the recommendations provided clear guidelines for assessing TILs, including the definition of specific parameters, implementing scoring systems, and establishing criteria for reporting. The guidelines emphasize sTILs as the principal parameter in future studies. It is recommended to evaluate them as an average, continuous parameter in one section under 200–400× magnification of invasive cancer, without focusing on hotspots. Additionally, previous biopsy sites, areas of necrosis, and areas with crush artifacts should be excluded. As sTILs are considered to be a more reproducible parameter than iTILs, which is due to the fact that sTILs are more abundant and more visible on H&E slides, the International TILs Working Group recommends evaluating sTILs [1]. Other parameters such as iTILs may be evaluated optionally to determine their significance.

## 2. Triple-Negative Breast Cancer (TNBC)

TNBC stands out among other BCs due to its occurrence in younger women, higher grade, aggressive clinical course, frequent lymph node and distant metastases, and significant mortality [13]. On the other hand, it is considered to have the highest immunogenic potential and abundance of TILs of all subtypes of BC. The majority of studies on the TNBC subtype showed average values of sTILs ranging from 15% to 25% of the tumor area [12].

TNBC frequently displays the morphology of invasive BCs with medullary features and lymphocyte-predominant BC (LPBC). The first one is characterized by a syncytial growth pattern, pushing margins and high TILs. The latter is defined as BC with sTILs ≥ 50–60%, which most often is of higher grade, has a triple-negative phenotype, or shows HER2 amplification/overexpression [14]. Although these cancers are high grade, they tend to have better prognosis than other cases of TNBC. One of the factors influencing this phenomenon is the presence of prominent TILs [1,15].

The potential prognostic value of sTILs in TNBC has been investigated across all stages. Loi et al. performed pooled data analysis of 2148 patients from nine studies in early TNBC treated with upfront surgery followed by AChT [16]. In total, 55.8% of patients received anthracycline-based schedules and 44.2% anthracycline with taxane. The average amount of sTILs was 23%. For all endpoints, invasive disease-free survival (iDFS), distant disease-free survival (dDFS), and OS, an increase in sTILs was associated with an improved prognosis, with HR of 0.87, 0.83, and 0.84 for every 10% of TILs increase, respectively. Moreover, in patients without lymph node metastases with sTILs ≥ 30% compared to patients with sTILs < 30% 3-year iDFS was 92% vs. 88%, 3-year dDFS was 97% vs. 91%, and 3-year OS was 99% vs. 95%. Based on these results, a prognostic model was generated to help estimate survival by incorporating the sTILs (available at www.tilsinbreastcancer.org accessed on 25 March 2024).

The role of TILs in another well-established treatment strategy for early TNBC, the combination of neoadjuvant chemotherapy (NAChT) with subsequent surgery, was analyzed in six studies performed by the German Breast Cancer Group [17]. In these studies, 906 out of 3771 (25%) pre-therapeutic core biopsies from primary BCs eligible for TIL assessment were TNBC. Tumors were divided into three predefined groups: low (0–10%), intermediate (11–59%), and high (≥60%) sTILs. Higher sTILs predicted better response to NAChT: 31% of patients with low sTILs, 31% with intermediate sTILs, and 50% with high sTILs achieved pCR. Additionally, in the univariate analysis, higher sTILs correlated with longer DFS, with an HR of 0.93, and OS, with an HR of 0.92, for each 10% increment [17]. In another study, concerning 134 patients with stage I-III TNBC who achieved pCR after NAChT, all those who had baseline TILs > 20% had better 5-year relapse-free survival (RFS) and 5-year OS (vs. 82.6% and 90.1% for those with fewer TILs) [18].

Studies on cohorts treated with neoadjuvant chemo-immunotherapy studies also demonstrated a positive relationship between baseline sTILs levels and tumor response, with pCR rates exceeding 70% noted in patients with high sTILs [19,20]. In a recent NeoPACT phase 2 study, which enrolled 117 stage I-III TNBC patients treated with six cycles of carboplatin + docetaxel + pembrolizumab, patients with sTILs ≥ 30% achieved a pCR rate of 78% vs. 45% in patients with sTILs < 30% [19]. Similar results were reported by the GeparNuevo study, including 174 patients with primary non-metastatic TNBC treated with neoadjuvant nab-paclitaxel followed by dose-dense epirubicin/cyclophosphamide, all combined with durvalumab or placebo, which showed that patients with intermediate/high TILs (≥11%) in primary sample had better iDFS than patients with low TILs (<11%) [21].

Notably, sTILs can potentially identify a subset of TNBC patients with an excellent prognosis even without AChT/NAChT. A recent study investigated 476 early-stage TNBC patients, where only surgery and/or radiation therapy was used without perioperative ChT. In stage I patients with sTILs ≥ 30%, 5-year iDFS was 91%, dDFS was 97%, and OS was 98%, while in the whole cohort, an increase in sTILs by every 10% was an independent, favorable prognostic factor for iDFS, dDFS, and OS with HRs of 0.93, 0.86, and 0.88, respectively [22]. Since the median age in this study was 64 years, which is significantly higher than typically seen in TNBC, these data may be useful in the context of determining indications for AChT in patients with stage I TNBC and comorbidities and/or age-related frailty, where potential benefit from chemotherapy may be counterbalanced by toxicity [23]. More informative data on younger patients are provided by a study from the Netherlands on the prognostic value of sTILs in node-negative TNBC patients < 40 years who did not receive perioperative ChT. In total, 441 patients were divided into three groups: low (<30%), intermediate (30–75%), and high (>75%) sTILs. The high sTILs group had a 15-year cumulative risk of distant metastases or death of 2.1%, irrespective of primary tumor size, in contrast to the low sTILs group risk of 38.4%. Moreover, every 10% increase in sTILs was associated with an improvement in OS, with HR 0.82 [23]. Finally, in an analysis of 1041 patients with stage I TNBC who were not treated with perioperative ChT, those with sTILs ≥ 30% had excellent 10-year breast cancer-specific survival (BCSS) (96% vs. 87% in patients with sTILs < 30%) [24].

Nonetheless, current guidelines recommend ChT for most patients with stage I TNBC, irrespective of TIL status. The European Society for Medical Oncology guidelines recommends AChT for ≥6 mm tumors [25], while The National Comprehensive Cancer Network (NCCN) guidelines recommend routine ChT for tumors ≥ 11 mm and case-by-case consideration for 6–10 mm tumors [26]. Following the promising results of the retrospective studies, trials are currently planned to assess the possibility of avoiding ChT in TILs-high early TNBC. TILs and their use as a prognostic factor in early BC have been endorsed by the St Gallen Consensus since 2019. However, for now, TIL scoring should not be used to support treatment decisions nor to escalate or de-escalate treatment [27]. The 2023 St Gallen Consensus expert panel specifically rejected the notion that a high TIL score should prompt the omission of ChT [28].

In metastatic TNBC (mTNBC) sTIL levels can identify patients with a greater chance of achieving response to immunotherapy, as demonstrated in the KEYNOTE-086 study of 228 patients with mTNBC with any PD-L1 expression who were administered single-agent pembrolizumab. Patients with sTILs ≥ 10% had better objective response rates (ORRs) than patients with sTILs < 10% (18.6% vs. 6.1%, respectively) [29] (Figure 1).

## 3. Human Epidermal Growth Factor Receptor 2 Positive (HER2+) BC

While TNBC and HER2+ BC are two distinct molecular subtypes of BC, they share certain similarities in clinicopathological characteristics and treatment challenges, such as high histologic grade, more aggressive biology, early metastatic potential, and higher rates of pCR with neoadjuvant treatment, compared to hormone receptor-positive BCs [30,31,32]. Both TNBC and HER2+ BC also exhibit higher levels of TILs compared to other BC subtypes.

One of the approaches employed in the early HER2-positive BC involves upfront surgery, followed by subsequent AChT combined with trastuzumab. The role of sTILs in this scenario has been studied in the ShortHER trial, which randomly assigned 1253 individuals diagnosed with early HER2+ BC to a combination of anthracycline and taxane-based ChT along with one year (long arm) or nine weeks (short arm) of trastuzumab [33]. sTILs low/high cut-off value was established at 20%. In the low sTILs group, there was a significant difference in 5-year dDFS between the long and short arm (93.3% vs. 88.8%, respectively), whereas, in the high sTILs group, 5 year dDFS was similar and very good in both arms (97.6% for short and 93.7% for long trastuzumab) [33]. In other words, in the low sTILs group, patients benefited from 1-year trastuzumab compared to 9-week treatment, whereas in the high sTILs group, outcomes were excellent regardless of trastuzumab duration. In a meta-analysis of 4097 early HER2+ BCs from randomized trials of trastuzumab + AChT, every 10% increase in TILs was associated with an approximately 14% lower risk of recurrence (*p* < 0.0001), regardless of whether patients received trastuzumab or not [34]. Thus, sTILs may refine the ability to identify patients eligible for treatment de-escalation, should this finding be confirmed in prospective studies. High sTILs also correlated with longer DFS and OS with anthracycline-only, but not anthracycline + taxane ChT, in a group of 297 HER2+ BC patients in BIG 02-98 trial; however, due to lack of anti-HER2 therapy use, these results may not be applicable to current patient populations [32].

In the meta-analysis involving a total of 9145 patients treated in the neoadjuvant setting, higher TILs in HER2+ BC were associated with larger benefits from NAChT combined with anti-HER2 therapy, expressed as higher pCR rate, longer DFS, and longer OS [35]. Another meta-analysis including five RCTs (CherLOB, GeparQuattro, GeparQuinto, GeparSixto, and NeoALTTO) confirmed a positive correlation between high TILs in treatment-naïve samples and pCR. Interestingly, this association was not affected by the addition of an anti-HER2 agent [36]. In another meta-analysis including 15 studies of HER2+ BCs treated with NAChT and 5 treated with AChT, He et al. identified a positive correlation of each 10% increase in TILs with pCR, with pooled OR of 1.27 and OS with pooled OR of 0.92 [37]. 

Another noteworthy issue is the shift in the percentage of TILs during NAChT. In the first report on this subject among 175 HER2+ BC patients receiving NAChT +/− trastuzumab, the baseline median TIL level of 25% (range 2–70%) after NAChT dropped to a median of 10%. The greater decrease between the initial number of TILs and the amount of lymphocytic infiltration in the tumor bed area after treatment was strongly associated with pCR, whereas patients who did not achieve pCR and had a post-NAChT TILs level > 25% demonstrated shorter DFS than those with lower TILs [9]. This implies that a greater abundance of TILs following NAChT is linked to a reduced likelihood of achieving pCR, and among patients failing to achieve pCR, it indicates a poorer prognosis.

Interestingly, the relationship between TILs and the effectiveness of NAChT combined with dual anti-HER2 blockade (trastuzumab with pertuzumab) seems to differ from that observed for anti-HER2 monotherapy. In the NeoSphere trial of neoadjuvant docetaxel combined with trastuzumab (TH), pertuzumab (TP), both (THP), or monoclonal antibodies alone (HP), TILs, as a continuous variable, were not associated with pCR [38]. Similarly, in the Tryphaena trial that randomly assigned 225 patients to trastuzumab/pertuzumab combined with anthracycline-based or anthracycline-free ChT, TILs were not associated with pCR rate; however, after a median of 4.7 years follow-up, each increment of 10% of baseline TILs was correlated with 25% reduction in event-free survival events [39].

The achievement of pCR in HER2+ BC may also be influenced by the cellular composition of TILs [40]. FOXP3+ regulatory T cells may suppress antitumor immunity, in contrast to CD8+ and CD4+ T cells, which play a role in promoting immune responses [41]. Surprisingly, in the analysis of 28 papers, HER2+ BC FOXP3+ cell-rich TILs were correlated with more frequent pCR (OR 1.20) and longer OS (HR 0.22). On the contrary, in TNBC and luminal BC, there was no significant correlation between the amount of FOXP3 cells and OS or pCR [40].

The significance of TILs as prognostic or predictive markers in advanced HER2+ BC has been scarcely documented. In the CLEOPATRA study, which established the combination of docetaxel with dual anti-HER2 blockade (trastuzumab and pertuzumab) as the standard first-line treatment for advanced HER2+ BC, higher TILs were associated with better OS [42]. In a substudy of 678 patients assessed at median follow-up for OS of 51 months, the mean sTILs value was 10%. In both study arms, an increase in sTILs by every 10% was associated with OS prolongation with adjusted HR of 0.89. Moreover, longer OS was demonstrated in patients with TILs > 20% vs. ≤20%. Conversely, in the MA.31 study, where a first-line combination of taxane with trastuzumb or lapatinib was used, the overall sTIL counts of below vs. above 5% did not show a significant effect on progression-free survival (PFS) (HR 1.04) [43]. One of the possible explanations of these conflicting results may be different sTILs cut-off points: for MA.31, TILs > 5% were considered high, while CLEOPATRA used the cut-off point of 20%; thus further research is needed on optimal TILs cut-off for metastatic HER2+ BC (Figure 2).

## 4. Luminal BC

Luminal BC, encompassing subtypes Luminal A and Luminal B, constitutes the most common group of BCs [44]. It is generally characterized by low levels of TILs, typically <10% [12]. Luminal A BC is characterized by the presence of low Ki67 (<20%), lower grade, more indolent clinical course, good response to HT, lower risk of relapse, and better prognosis [44]. Luminal B cancers show higher Ki67, more aggressive clinical course, more frequent distant metastases, and a greater number of TILs [45]. Studies evaluating the correlation between the number of TILs and survival in patients with luminal BC receiving AChT are inconsistent with relationships observed for TNBC and HER2+ BC [32,46]. The analysis of 3771 BCs showed a correlation between high baseline TILs and pCR in all BC subtypes; however, in luminal BCs, better OS was paradoxically associated with lower TIL scores [17]. Different conclusions were obtained in the meta-analysis of 33 studies in which high TILs in luminal BC were not associated with a higher pCR rate, yet again, high TILs were correlated with shorter OS [47]. In contrast, a group of 344 patients with high-risk early ER+/HER2− BC treated with nivolumab + NAChT patients with ≥1% sTILs had a greater rate of pCR than patients with sTILs < 1% [48]. Worse prognosis in luminal cancers with more TILs may result from a higher grade, higher Ki67, and different cellular composition of TILs in this BC subtype, in particular a higher amount of FOXP3+ cells not accompanied by a higher number of CD8+ cells [49,50]. Hence, it seems that high TILs may be a reflection of a more aggressive phenotype in luminal BC, rather than an independent prognostic factor.

Single hormone receptor-positive BCs characterized by the expression of only one hormone receptor (ER or PR), albeit classified as luminal-like, exhibit distinct biological features and clinical behaviors [51]. Studies evaluating the number of TILs and their prognostic significance in single hormone receptor-positive BC are scarce. In a study from our group, which evaluated 197 patients with single hormone receptor-positive BC (121 ER+/PR− and 76 ER−/PR+), ER−/PR+ BCs were characterized by a significantly higher number of sTILs than ER+/PR− BCs. Additionally, in the whole cohort, patients with low sTILs (<10%) had a higher risk of death [52] (Figure 3).

## 5. Ductal Carcinoma In Situ (DCIS)

DCIS, in spite of being a preinvasive tumor, may still exhibit aggressive clinical behavior, with a significant incidence of local relapses, including invasive disease. Thus, better identification of prognostic factors in this malignancy is urgently needed and the presence of TILs within the tumor microenvironment has been the focus of research in recent years. The method of TILs assessment in DCIS is not yet clearly established, leading to challenges in achieving reproducibility of results and understanding their prognostic significance in this particular setting [53]. The major constituents of TILs in DCIS in decreasing proportions are CD3+ T cells, CD8+ T cells, CD20+ B cells, and FOXP3+ regulatory T-cells [53]. The impact of TILs density and the composition of TILs on the prognosis in DCIS based on available studies is ambiguous. In a study of 534 DCIS cases, dense TILs accompanied DCIS of larger size, with comedo necrosis, of intermediate and high grade, with concomitant Paget’s disease and lack of estrogen receptor expression, occurring at younger age and were associated with shorter recurrence-free interval [54]. A similar correlation was obtained in the group of 283 DCIS cases in which patients with TILs > 17% had a higher risk of recurrence [55]. In another study of 1488 DCIS patients, higher TILs correlated with HER2+ phenotype, higher grade, and necrosis, with no impact on ipsilateral DCIS or invasive tumor recurrence, irrespective of the treatment method [56]. However, in a meta-analysis of seven studies including 3437 DCIS cases, high TILs were associated with triple-negative and HER2+ phenotype, high grade and the presence of necrosis, as well as a higher risk of invasive and non-invasive recurrence [57] (Figure 4).

## 6. Conclusions

What distinguishes our work from previous reports is the emphasis on the significance of TILs not only in the context of biological subtypes of BC but also in relation to particular therapeutic settings within each biological subtype of BC. We also incorporated data for DCIS, a topic omitted in prior studies. We aimed to highlight the role of TILs in the dialogue between pathologists and oncologists in light of the latest findings from clinical research and recommendations from scientific societies, including data from the most recent scientific meetings.

In conclusion, TILs hold considerable prognostic and predictive value in BC, with variations observed across different BC phenotypes. The most compelling evidence supporting the clinical significance of TILs is found in TNBC. Understanding the intricate relationship between TILs infiltration and prognosis in specific BC subtypes is crucial for the development of effective treatment strategies. Moreover, TILs have the potential to serve as predictive biomarkers, possibly guiding treatment decisions and optimizing patient outcomes.

Despite the compelling evidence and existence of well-defined standards of evaluation, in many of the available papers, the assessment of TILs in BC still lacks standardization. The methods used in individual studies to assess TILs differed significantly. Some of them evaluated TILs in core needle biopsies, some on full sections, and for some such information was not provided. In some studies, TILs were assessed in accordance with the recommendations provided by an International TILs Working Group 2014 and others adopted their own assessment algorithm or used an algorithm proposed in another study. There is a particular difficulty with the assessment of TILs in meta-analyses because the studies included in them used different cut-off values. Furthermore, not all studies describe the detailed method for assessing TILs. Additionally, TIL scoring mostly relies on subjective assessment by pathologists, introducing the possibility of interobserver variability. This lack of uniformity presents challenges in comparing results across different studies and hampers the establishment of consistent associations between TILs and clinical outcomes. By addressing these methodological limitations and promoting consistency in TIL assessment, the full potential of TILs as valuable biomarkers in BC may be unlocked, aiding in treatment decision-making and improving patient care (Table 1).

## Figures and Tables

**Figure 1 biomedicines-12-00763-f001:**
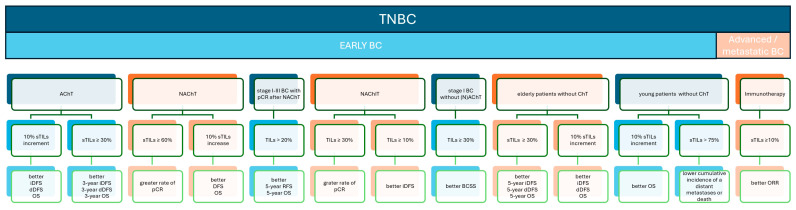
TILs as biomarkers in TNBC: prognostic and outcome graphical summary. Study ref No. [16,17,18,19,21,22,23,24,29].

**Figure 2 biomedicines-12-00763-f002:**
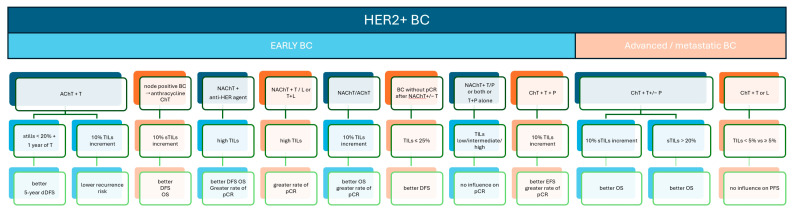
TILs as biomarkers in HER2+BC: prognostic and outcome graphical summary. Study ref No. [9,32,33,34,35,36,37,38,39,42,43].

**Figure 3 biomedicines-12-00763-f003:**
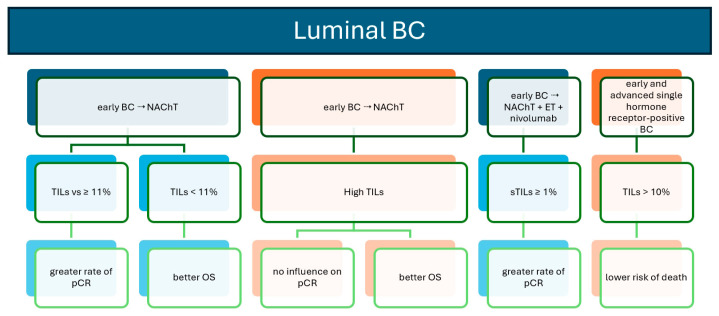
TILs as biomarkers in luminal BC: prognostic and outcome graphical summary. Study ref No. [17,47,48,52].

**Figure 4 biomedicines-12-00763-f004:**
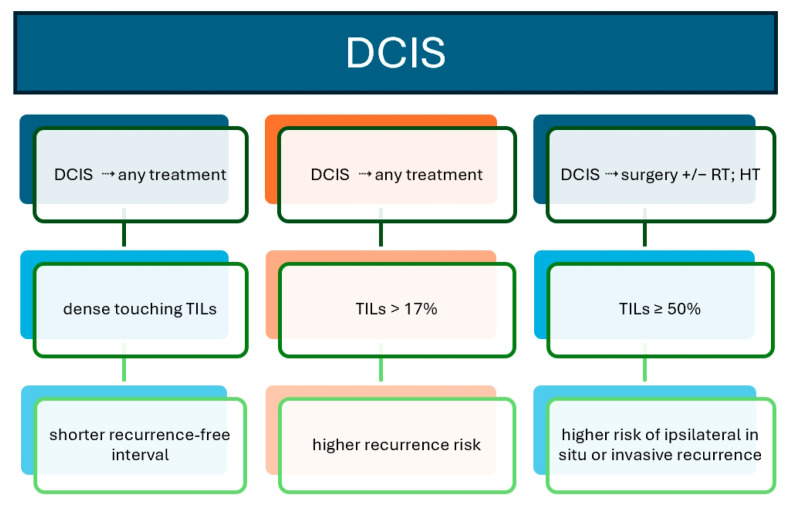
TILs as biomarkers in DCIS: prognostic and outcome graphical summary. Study ref No. [54,55,57].

**Table 1 biomedicines-12-00763-t001:** TILs as biomarkers in breast cancer: subtype-specific prognostic and outcome measures.

Subtype	Setting	Setting/Population Details	N	Subgroup	Outcome Measure	Outcomes		*p* Value	Study Ref. No.
TNBC	early BC	cancer treated with AChT	2148	10% sTILs increment	iDFS	HR 0.87 (95% CI: 0.83 to 0.91)		<0.000001	[16]
					dDFS	HR 0.83 (95% CI: 0.79 to 0.88)		<0.000001	
					OS	HR 0.84 (95% CI: 0.79 to 0.89)		<0.000001	
		node-negative cancer treated with AChT	706	sTILs ≥ 30% vs. <30%	3-year iDFS	92% (95% CI: 0.89–0.96) vs. 88% (95% CI: 0.85–0.90)		<0.0001	
					3-year dDFS	97% (95% CI: 0.95–0.99) vs. 91% (95% CI: 0.88–0.83)		<0.0001	
					3-year OS	99% (95% CI: 0.97–1.00) vs. 95% (95% CI: 0.93–0.97)		<0.0001	
		cancer treated with NAChT	906	sTILs < 10%	pCR	31%		<0.0001	[17]
				sTILs 11–59%		31%			
				sTILs ≥ 60%		50%			
				10% sTILs increment	DFS	HR 0.93 (95% CI: 0.87–0.98)		0.011	
					OS	HR 0.92 (95% CI: (0.86–0.99)		0.032	
		stage I–III treated with NAChT	134	TILs > 20% vs. <20%	5-year RFS	100% vs. 82.6%		<0.001	[18]
					5-year OS	100% vs. 90.1%		<0.007	
		cancer treated with NAChIT	117	sTILs ≥ 30% vs. <30%	pCR	78% vs. 45% (OR: 4.39; 95% CI: 1.63–11.82)		0.003	[19]
			174	TILs ≥ 11% vs. <11%	iDFS	HR 0.55 (95% CI: 0.28–1.07)		0.0079	[21]
		stage I without (N)AChT	1041	sTILs ≥ 30% vs. <30%	BCSS	96% vs. 87% HR 0.45 (95% CI: 0.26–0.77)		ND	[24]
		stage I cancer in elderly patients treated without ChT	74	≥30% sTILs	5-year iDFS	91% (95% CI: 84% to 96%)		ND	[22]
					5-year dDFS	97% (95% CI: 93% to 100%)			
					5-year OS	98% (95% CI: 95% to 100%)			
		cancer in elderly patients treated without ChT	476	10% sTILs increment	iDFS	HR 0.93 (95% CI: 0.82–0.97)		0.012	
					dDFS	HR 0.86 (95% CI: 0.77–0.95)		<0.01	
					OS	HR 0.88 (95% CI: 0.79–0.98)		0.015	
		cancer in young patients treated without ChT	441	10% sTILs increment	OS	HR 0.82 (95%CI 95% CI: 0.77–0.88)		<0.001	[23]
				sTILs < 30%	cumulative incidence of a distant metastases or death	38.4% (95% CI: 32.1 to 44.6)		ND	
				sTILs > 75%		2.1% (95% CI: 0 to 5.0)			
	advanced/metastatic BC	Immunotherapy in metastatic BC	228	sTILs ≥ 10% vs. <10%	ORR	18.6% vs. 6.1%		0.012	[29]
HER2+	early BC	cancer treated with AChT + T	866	sTILs < 20%; 9 weeks of T vs. sTILs < 20%; 1 year of T	5-year dDFS	88.8% vs. 93.3% (HR 1.75, 95% CI: 1.09–2.8)		0.02	[33]
				sTILs ≥ 20%; 9 weeks of T vs. sTILs ≥ 20%; 1 year of T		97.6% vs. 93.7% (HR 0.23, 95% CI: 0.05–1.09)		0.064	
		cancer treated with AChT + T	4097	10% TILs increment	recurrence risk	HR 0.87 (95%CI: 0.84–0.9)		<0.0001	[34]
		node positive cancer treated with anthracycline ChT	297	10% sTILs increment	DFS	ND		0.042	[32]
					OS	ND		0.018	
		cancer treated with NAChT + anti-HER agent	9145	high TILs vs. low TILs	pCR	pooled OR 2.19 (95% CI: 1.06–4.52)		0.035	[35]
					DFS	pooled HR 0.95 (95% CI: 0.92–0.98)		0.0003	
					OS	pooled HR 0.93 (95% CI: 0.87–0.99)		0.01	
		cancer treated with NAChT + T, L or T + L	1256	high TILs vs. low TILs	pCR	OR 2.46 (95% CI: 1.36–4.43)		0.035	[36]
		cancer treated with NAChT/AChT	1801	10% TILs increment	pCR	pooled OR 1.27 (95% CI: 1.19–1.35)		0.01	[37]
			1985		OS	HR 0.92 (95% CI: 0.89–0.95)		0.06	
		cancer treated with NAChT +/− T	107	TILs ≤ 25% vs. >25%	DFS	HR 3.23 (95% CI: 1.05–9.93)		0.03	[9]
		cancer treated with NAChT + T/P or both or T + P alone	243	TILs < 5%	pCR	TH	0%	TH 0.157 THP 0.240 HP 0.413 TP 0.685 combined 0.062	[38]
						THP	28.6%		
						HP	0%		
						TP	12.5%		
						combined TH, HP, TP	4.3%		
				TILs 5–49%		TH	36.4%		
						THP	48.9%		
						HP	19.9%		
						TP	25%		
						combined TH, HP, TP	26.9%		
				TILs ≥ 50%		TH	33.3%		
						THP	22.2%		
						HP	20%		
						TP	28.6%		
						combined TH, HP, TP	26.7%		
		cancer treated with ChT + T + P	213	10% TILs increment	pCR	OR: 1.12 (95% CI: 0.95–1.31)		0.17	[39]
					EFS	adjusted OR: 0.75 (95% CI: 0.56 –1.00)		0.05	
	advanced/metastatic BC	cancer treated with ChT + T +/− P	678	10% sTILs increment	OS	adjusted HR 0.89 (95% CI: 0.83–0.96)		0.0014	[42]
				sTILs > 20% vs. ≤20%		HR 0.76, 95%CI: 0.6–0.96		0.021	
		cancer treated with ChT+ T or L	614	sTILs < 5% vs. ≥5%	PFS	HR 1.04 (95% CI: 0.84–1.28)		0.74	[43]
Luminal BC	early BC	cancer treated with NAChT	1366	TILs < 11%	pCR	6%	OR 1.31 (95% CI: 1.23–1.41)	<0.0001	[17]
				TILs 11–59%		11%			
				TILs ≥ 60%		28%			
			832	TILs < 11% vs. ≥11%	OS	HR 1.1 (95% CI: 1.02–1.19)		0.011	
			1597	high TILs vs. low TILs	pCR	OR: 1.154 (95% CI: 0.789 –1.690)		0.46	[47]
			1829		OS	HR 1.077 (95% CI: 1.016 –1.141)		0.012	
		cancer treated with NAChT and ET + nivolumab	344	sTILs ≥ 1% vs. <1%	pCR	ND		ND	[48]
	early and advanced BC	single hormone receptor-positive BC	197	TILs > 10% vs. <10%	risk of death	HR 3.14 (95% CI: 1.37–7.19)		0.006	[52]
DCIS		carcinoma in situ treated with any treatment	534	dense touching TILs vs. sparse touching TILs	recurrence-free interval	HR 2.573 (95% CI: 1.412–4.69)		0.002	[54]
		carcinoma in situ treated with any treatment	283	TILs > 17% vs. <17%	recurrence risk	HR 2.97 (95% CI: 1.17–7.51)		0.02	[55]
		carcinoma in situ treated with surgery +/− RT; HT	2941	TILs ≥ 50% vs. <50%	ipsilateral in situ or invasive recurrence	OR: 2.05 (95% CI: 1.03–4.08)		0.402	[57]

Legend: AChT—adjuvant chemotherapy; BCSS—breast cancer-specific survival; ChT—chemotherapy; HT—hormonotherapy; L—lapatinib; NAChT—neoadjuvant chemotherapy; NAChIT—neoadjuvant chemo-immuno therapy; ND—no data; ORR—objective response rate; P—pertuzumab; RFS—relapse-free survival; RT—radiotherapy; T—trastuzumab.

## Data Availability

All data underlying the results are available as part of the article and no additional source data are required.

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
