# Peer review of "Tumor-Infiltrating Lymphocytes (TILs) in Breast Cancer: Prognostic and Predictive Significance across Molecular Subtypes"

_biomedicines, 2024, doi:10.3390/biomedicines12040763_

Round 1

Reviewer 1 Report

Comments and Suggestions for Authors

In this review, the authors discussed how tumor-infiltrating lymphocytes may predict breast cancer prognosis, based on previous clinical and molecular works. They focus on multiple BC subtypes, particularly triple-negative, HER2+, luminal, and ductal carcinomas. The text is comprehensive but needs improvement for clarity. We suggest that the authors introduce the different BC subtypes briefly before focusing on the specifics of TIL-related results. Other suggestions are described below. 

Abstract: please check the break at line 19/20.

When introducing the BC subtypes, we suggest broadening the literature review to mention other topics besides TIL abundance in different BC subtypes, as this is not their most defining characteristic, but one of them.

Line 85-87: Better prognosis in invasive TNBC subtypes depends on multiple variables, including but not limited to TIL presence. Please revise the paragraph to avoid generalizations.

Check that punctuation is placed after the references - some examples are found in lines 87, 90, 104, 114, and so on. References should also be truncated whenever possible - check line 169.

We suggest the display of data with multiple numbers (i.e., values in %, CI, p values, etc) to be exclusive to tables, leaving only the most important values for discussion in the text. E.g. lines 97-99, 130-134, etc.

Please correct the double spacing in lines 161 and 245.

Comments on the Quality of English Language

We suggest the authors check the appropriate placement of the plural for TIL (i.e., TILs) as not all applications on the text are correct. We also suggest that a thorough English correction is performed to avoid overly long sentences and dubious meanings, as observed in lines 68-72, 81-86, etc. 

Reviewer 2 Report

Comments and Suggestions for Authors

The article titled "Tumor-Infiltrating Lymphocytes (TILs) in Breast Cancer: Prognostic and Predictive Significance Across Molecular Subtypes" by Aleksandra Ciarka et al. offers valuable insights. However, there are several areas that require attention:

1. The authors should provide their own justification for the study, as previous publications have already explored the relevance of the topic. Examples of such publications include articles in PubMed, such as Cancers (Basel). 2023 Jan 26;15(3):767. doi: 10.3390/cancers15030767; Cancers (Basel). 2023 Sep 8;15(18):4479. doi: 10.3390/cancers15184479; BMC Cancer. 2020 Nov 25;20(1):1150. doi: 10.1186/s12885-020-07654-y; Lancet Oncol. 2018 Jan;19(1):40-50. doi: 10.1016/S1470-2045(17)30904-X; Clin Transl Oncol. 2016 May;18(5):497-506. doi: 10.1007/s12094-015-1391-y; PLoS One. 2016 Apr 13;11(4):e0152500. doi: 10.1371/journal.pone.0152500; Diagn Pathol. 2022 Nov 21;17(1):91. doi: 10.1186/s13000-022-01271-y, among others. Consequently, the study does not provide any innovative information.

2. The authors should provide more explicit details about the methods used to assess TILs and the challenges associated with standardizing these assessment techniques.

3. The title of Table 1 should be modified to "TILs as Biomarkers in Breast Cancer: Subtype-Specific Prognostic and Outcome Measures."

4. The authors could consider incorporating figures that depict the comprehensive study of TILs in Breast Cancer.

Comments on the Quality of English Language

Moderate editing of English language required

Reviewer 3 Report

Comments and Suggestions for Authors

Manuscript entitled "Tumor-Infiltrating Lymphocytes (TILs) in Breast Cancer: Prognostic and Predictive Significance Across Molecular Subtypes"

This work is not well-organized and not easily read. I would suggest the following modifications:

1. The table is not well formed. The authors should provide figures to better illustrate these issues.

2. A summary figure should be provided to better adress the story.

3. The authors, instead of just mention the TILs count, should also provide a detail review on the molecular mechanism.

Comments on the Quality of English Language

Acceptable

Round 2

Reviewer 1 Report

Comments and Suggestions for Authors

The authors have corrected the major issues and the manuscript is now ready for publication.

Reviewer 2 Report

Comments and Suggestions for Authors

 Accept in present form

Comments on the Quality of English Language

Minor editing of English language required

Reviewer 3 Report

Comments and Suggestions for Authors

This work is well revised and is acceptable in the present form.

Comments on the Quality of English Language

Acceptable